# Enhancing Generalization of First-Order Meta-Learning

**Mirantha Jayathilaka**
Department of Computer Science
University of Manchester
Manchester, UK
`mirantha.jayathilaka@manchester.ac.uk`

## Abstract

In this study we focus on first-order meta-learning algorithms that aim to learn a parameter initialization of a network which can quickly adapt to new concepts, given a few examples. We investigate two approaches to enhance generalization and speed of learning of such algorithms, particularly expanding on the Reptile (Nichol et al., 2018) algorithm. We introduce a novel regularization technique called meta-step gradient pruning and also investigate the effects of increasing the depth of network architectures in first-order meta-learning. We present an empirical evaluation of both approaches, where we match benchmark few-shot image classification results with 10 times fewer iterations using Mini-ImageNet dataset and with the use of deeper networks, we attain accuracies that surpass the current benchmarks of few-shot image classification using Omniglot dataset.

## 1 Introduction

A common drawback consistently seen in traditional machine learning algorithms is the need for large amounts of training data in order to learn a given task (Lake et al., 2015), whereas the ability to grasp new concepts with just a few examples is clearly seen in the way people learn (Lake et al., 2017). This offers many challenges in fast adaption of machine learning in new fields and hence there is a growing interest in algorithms that can learn with limited data availability (Santoro et al., 2016).

In the development of learning methods that can be trained effectively on sparse data, the process of learning-to-learn is seen as a crucial step (Andrychowicz et al., 2016). This is often termed as meta-learning (Schaul & Schmidhuber, 2010), where a variety of techniques have been presented. In our study, we specifically focus on approaches that learn an initialization of a network, trained on a dataset of tasks. Model-agnostic meta-learning (MAML) (Finn et al., 2017) presented this exact approach and its applications of few-shot image classification, where a task was defined as correct classification of a test image out of $N$ object classes, after training on a set of $K$ examples per each class. Furthermore, MAML presented its first-order variant, where the second order derivatives were eliminated during computation while preserving results of the benchmarks. The approach avoided the computational expense of second order derivatives by treating them as constants. First-order meta-learning was further investigated in the Reptile algorithm (Nichol et al., 2018), where the implementation was simplified eliminating the need for a test set in the tasks. Our study uses Reptile as the algorithm of choice to incorporate the techniques presented to improve generalization of first-order meta-learning.

Even though first-order meta-learning has shown to attain fast generalization of concepts given limited data, empirical evaluations on few-shot image classification tasks (Nichol et al., 2018) show potential to improve the outcomes, especially on inputs with richer features such as real world images. Also drawbacks are seen in slower convergence, requiring a large number of iterations during the training phase. In this study, we investigate techniques used to obtain higher task generalization in models such as regularization (Srivastava et al., 2014) and deeper networks (Krizhevsky et al., 2012) and we present ways of adapting those in first-order meta-learning.

The contributions of our study are as follows.

- Introduction of meta-step gradient pruning, a novel approach to regularize parameter updates in first-order meta-learning.
- Empirical evaluation of meta-step gradient pruning, achieving benchmark few-shot image classification accuracies with 10 times fewer iterations.
- Empirical evaluation of deeper networks in the meta-learning setting, achieving results that surpass the current benchmarks in few-shot image classification.

## 2 FIRST-ORDER META-LEARNING

The goal of the meta-learning algorithm investigated in this study, is to learn the parameter initialization of a network, so that it can be adapted to generalizing new unseen data points with just few examples during training. These initialization parameters are optimized by training on a dataset of tasks. A task is sampled to be a set of $N$ classes with $K$ training examples per each class (Ravi & Larochelle, 2016). In this section we will formalize the algorithm and in Section 3, our approaches to enhance generalization are described.

In order to understand the first-order aspect of the algorithm, we will first look at the approach of MAML (Finn et al., 2017). Compute a set of initial parameters $\theta$, so that the loss $L_{\tau_i}$ of a task drawn from the distribution of tasks $P(\tau)$ is minimized after $m$ iterations. This is shown in Equation 1.

$$minimize \ L_{\tau_i}(f_{\tau_i}^m(\theta)) \tag{1}$$

Here, $f_{\tau_i}^m(\theta)$ is the function that updates the initial parameters $\theta, m$ times on the task examples. In our experiments of few-shot image classification in Section 4, this function is optimized using the Adam (Kingma & Ba, 2014) algorithm. Further in MAML, Equation 1 is modified so that the loss $L_{\tau_i}$ is obtained on the test samples of the task, while the update function $f_{\tau_i}^m(\theta)$ uses the train samples. Considering a task of a few-shot image classification, this loss corresponds to the classification loss of test images of classes $N$, after training on $K$ samples of each class from the train set. Stochastic gradient decent is used to optimize on this loss in order to update the initial parameters. We call this the meta-step and it is done by computing the gradient of Equation 1 w.r.t. $\theta$ as shown in Equation 2.

$$\begin{aligned} gradient &= \frac{\partial}{\partial \theta} L_{\tau_i}(f_{\tau_i}^m(\theta)) \\ &= J_{\tau_i}'(\theta) L_{\tau_i}'(\tilde{\theta}) \end{aligned} \tag{2}$$

In Equation 2, $\tilde{\theta} = f_{\tau_i}^m(\theta)$, and $J_{\tau_i}'(\theta)$ is the Jacobian matrix of the function $f_{\tau_i}^m$. Here, we can interpret $f_{\tau_i}^m$ as the summation of gradient updates after $m$ iterations of training. The key idea of first-order meta-learning is treating these gradients as constants, so that the Jacobian matrix $J_{\tau_i}'(\theta)$ can be replaced with an identity operation. Hence, the outer-loop optimization on the initial parameters simply becomes $gradient = L_{\tau_i}'(\tilde{\theta})$ (Finn et al., 2017).

The Reptile (Nichol et al., 2018) algorithm further simplifies the above first-order update by replacing the gradient vector $L_{\tau_i}'(\tilde{\theta})$ to be the difference between the vectors of updated parameters and initial parameters $\tilde{\theta} - \theta$. We develop our generalization techniques on this algorithm in Section 3 and show empirical evaluation in Section 4

## 3 OUR APPROACH

We introduce a novel regularization method for first-order meta-learning known as meta-step gradient pruning. This is implemented by defining two new hyper-parameters $\gamma$ and $\psi$, that takes effect during the meta-step update of the training process. $\gamma$ defines a threshold value for $(\tilde{\theta} - \theta)$ difference in the meta-step and $\psi$ decides the number of iterations that gradient pruning will be effective, towards the end of the entire training process. Algorithm 1 shows the modified first-order meta-learning algorithm with meta-step gradient pruning. $\epsilon$ in the meta update denotes the meta-step size hyper-parameter.

---

**Algorithm 1** Reptile-based first-order meta-learning algorithm with meta-step gradient pruning

---
Initialize parameters $\theta$
Define hyper-parameters $\gamma, \psi, \epsilon$
**for** iterations = 1,2,...m **do**
    Sample a training task $\tau_i$ from $P(\tau)$ and compute $f_{\tau_i}^m(\theta)$ and loss $L_{\tau_i}$
    Updating model $m$ times using Adam to obtain $\tilde{\theta}$
    **if** current portion of executed iteration $< \psi$ **then**
        Update $\theta \longleftarrow \theta + \epsilon(\tilde{\theta} - \theta)$
    **else**
        Update $\theta \longleftarrow \theta + \epsilon(\tilde{\theta} - \theta)$ only if $|(\tilde{\theta} - \theta)| > \gamma$, otherwise $(\tilde{\theta} - \theta) = 0$
    **end if**
**end for**

---

The key intuition behind this approach is that, smaller meta-step updates towards the end of the training process tend to overfit on the train set tasks rather than generalizing. This was noticed from the considerable low test accuracies compared to train accuracies of the previous implementations. Our new addition of meta-step gradient pruning has helped to reduce this gap in both Omniglot and Mini-ImageNet few-shot image classifications tasks, along with fast convergence in 10 times less number of iterations than the previous implementations.

Secondly, we investigated the effects of introducing deeper networks in the inner-loop of $f_{\tau_i}^m(\theta)$ and evaluated the performance of the algorithm on the benchmark tasks. We introduced a 2 fold increase in depth for the same architecture used in MAML (Finn et al., 2017) implementation on Omniglot task, so that we could compare the results easily. We found that this approach gave a considerable increase in accuracy in the Omniglot image classification task. See results in Section 4.2.

## 4 EXPERIMENTS

The experiments carried out in this study used Omniglot (Lake et al., 2011) and Mini-ImageNet (Vinyals et al., 2016) datasets, which are often used as benchmarks in few-shot image classification tasks. Our results are shown in detail in Section 4.2.

### 4.1 FEW-SHOT CLASSIFICATION SETUP

We perform $K$-shot $N$-way classification tasks, where we sample $K$ number of examples per $N$ number of classes for a task. The network architectures and data preprocessing were implemented using the same methods as in MAML (Finn et al., 2017). Adam optimizer was used in the inner-loop while SGD in used for the meta-step update. The hyper-parameters $\gamma$ and $\psi$ from Algorithm 1 were tuned during training. We compare our results with both first-order MAML (Finn et al., 2017) and Reptile (Nichol et al., 2018) results obtained from the respective literature.

### 4.2 RESULTS

We first evaluated the effects of our technique of meta-step gradient pruning by computing the train and test accuracies produced during training of 1-shot 5-way image classification tasks using Omniglot and Mini-ImageNet datasets. The results are shown in Table 1. These results were produced after 5000 iterations on Omniglot dataset and 10000 iterations on Mini-ImageNet dataset, while keeping all other hyper-parameters the same. The reduced gap between the train and test accuracies imply the improved generalization of our approach compared to the Reptile implementation.

Few-shot image classification results on Mini-ImageNet dataset further produced some more promising results with the addition of meta-step gradient pruning. The outcomes are shown in Table 2 and we compare them with both MAML and Reptile algorithm results. We were able to almost match the benchmark performance on Mini-ImageNet dataset with 10 times fewer number of iterations. In other terms, this can be seen as a 10x increase in convergence speed. This shows that, enhanced generalization using meta-step gradient pruning produces noticeable increments in execution speeds.

Table 1: Comparing the gap between train and test set accuracies of Reptile and our approach in 1-shot 5-way few-shot image classification

| Dataset | Algorithm | Train accuracy | Test accuracy | Difference |
|---|---|---|---|---|
| Omniglot | Reptile | 85.26% | 83.03% | 2.23% |
| | Our algorithm | 86.81% | 85.27% | 1.54% |
| Mini-ImageNet | Reptile | 48.45% | 43.06% | 5.39% |
| | Our algorithm | 48.62% | 45.63% | 2.99% |

Table 2: Results on Mini-ImageNet few-shot classification with meta-step gradient pruning

| Algorithm | 1-shot 5-way | 5-shot 5-way |
|---|---|---|
| First-order MAML | 48.07% | 63.15% |
| Reptile | 47.07% | 62.74% |
| **Our algorithm (10 times fewer iterations)** | 46.89% | 61.94% |

Table 3: Results on Omniglot few-shot classification with deeper inner-loop networks. Best results are highlighted.

| Algorithm | 1-shot 5-way | 5-shot 5-way |
|---|---|---|
| First-order MAML | 98.30% | 99.20% |
| Reptile | 95.39% | 98.90% |
| Reptile + Transduction | 97.68% | 99.48% |
| Our approach (Deeper Reptile) | 97.12% | 99.02% |
| Our approach (Deeper Reptile + Transduction) | **98.82%** | **99.63%** |

Our approach of introducing deeper networks in the inner-loop clearly outperformed the previous implementation results of both first-order MAML and Reptile. We were able to set new benchmark results on Omniglot few-shot image classification tasks, in 1-shot 5-way and 5-shot 5-way settings. But weaknesses were seen in increased computations and time consumption. The results are shown in Table 3.

## 5 DISCUSSION

Our proposed novel approach of meta-step gradient pruning demonstrated enhanced generalization effects on the outcomes of first-order meta-learning. The reduced gaps between train and test set accuracies, during training of Omniglot and Mini-ImageNet few-shot classification tasks showed that the parameter initialization has learned to generalize better on the train set.

We were able to almost match the benchmark results of first-order MAML and Reptile implementations with 10 times fewer iterations using our algorithm. This further emphasized the improved generalization, helping the parameters to converge the loss on few-shot classification. This increase in speed is vital in tasks such as Mini-Imagenet, because performing first-order meta-learning on real world noisy images is computationally expensive and time-consuming.

With our approach of introducing deeper networks to the inner-loop in Omniglot few-shot classification, we showed results surpassing the current benchmarks of both first-order MAML and Reptile algorithms. The expanded parameter space with deeper models shows higher generalization as expected, but it makes the implementation more computationally expensive. This was identified as one drawback of this approach when applying to richer input data such as Mini-ImageNet tasks.

Enhanced and fast generalization is utmost important when learning with limited data. Looking forward, we see the importance of elaborated theoretical analysis of meta-step gradient pruning and more techniques of regularization during meta-learning. Also in the future we plan to investigate on the application of first-order meta-learning in other applications such as reinforcement learning.

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
