# OpenReview forum: "Enhancing Generalization of First-Order Meta-Learning"
_ICLR.cc/2019/Workshop/LLD — LLD 2019_

### Official Review · AnonReviewer1 · 2019-04-07
**A first step towards gaining a better understanding of first-order meta-learning**

**Rating:** 3
**Confidence:** 3

**Review:**

This work presents an empirical study of the first-order meta-learning Reptile algorithm, in particular investigating a proposed regularization technique and deeper networks. Their regularization method is to train as usual for the first \psi steps and subsequently only apply the training update to the learned initialization if the difference between that initialization before and after the task-specific update on the current task is greater than another hyperparameter \gamma.

They show experimentally that when training Reptile using the above procedure it overfits less: the gap between the training and testing accuracy is smaller (though not by a significant amount). What is perhaps more impressive is that they can obtain similar to state-of-the-art results on Omniglot and mini-ImageNet by applying 10x less updates than the corresponding state-of-the-art methods. Finally, they show that using deeper networks yields a benefit on Omniglot. This is an interesting observation, contradicting the intuition that when learning from little data larger networks would be more prone to overfitting.

Some concerns / suggestions:
-    I’m curious if a similar behaviour to the proposed regularization can be obtained simply by using a learning rate schedule, and / or using ADAM in the outer loop as well.
-    The results in Table 2 are slightly lower than MAML and Reptile’s and it’s not clear how many additional iterations would be required to match those results. It may be that to squeeze out that last bit of performance a lot more updates are required even with the proposed method. It therefore seems more informative to keep running the proposed method until that performance is reached and then compare the number of iterations required. Alternatively, showing the curve of the performance on held-out data throughout training would address this point as well.
-    Regarding the experiments with the deeper networks: the authors describe this as using deeper networks in the inner loop specifically. It found this confusing. Are the additional weights only “fast weights” (e.g. part of a task-specific classifier) that are not meta-learned (by the outer loop)? It would be useful to be more specific about this.
-    It would be interesting to present deeper network experiments on mini-ImageNet instead of (or in addition to) Omniglot, since it’s a more realistic and challenging benchmark with larger resolution images.
-    From what I understand, the proposed modifications are not applicable exclusively to first-order meta-learning. I would therefore be curious about whether applying these to second-order methods (e.g. full MAML) would yield similar conclusions.

Overall, I feel that meta-training is still poorly understood so I think empirical investigations like the one in this work are useful for gaining stronger intuitions for best practices in this setup. I therefore recommend acceptance of this work.

---

### Official Review · AnonReviewer2 · 2019-04-11
**This in an interesting simple extension, results seem promising, but very preliminary**

**Rating:** 3
**Confidence:** 2

**Review:**

The paper introduces a simple extension of first-order MAML/Reptile algorithm. It proposes to stop the inner loop if the magnitude of the update does not exceed a certain threshold. Exprerimentation looks promising, although confusing. It is not clear why Table 1 accuracies are different from Table 2. For Table 2 it makes sense to demonstrate learning curves to emphasize the convergence speed. Metalearning algorithms are known to have high variance, so it makes sense to report error bars across multiple seeds.

All in all, the idea to improve the convergence is worth exploring and interesting to discuss, but the paper is a bit raw.

---

### Decision · Program_Chairs · 2019-04-16
**Acceptance Decision**

Accept